# Impulsivity and Its Association with Depression and Anxiety in the Normal Egyptian Population Post COVID-19 Pandemic

**DOI:** 10.3390/medicina60081367

**Published:** 2024-08-22

**Authors:** Marwa S. Ismael, Marwa O. Elgendy, Ammena Y. Binsaleh, Asmaa Saleh, Mohamed E. A. Abdelrahim, Hasnaa Osama

**Affiliations:** 1Psychiatry Department, Faculty of Medicine, Beni-Suef University, Beni-Suef 62521, Egypt; marwa_safwat@med.bsu.edu.eg; 2Department of Clinical Pharmacy, Beni-Suef University Hospitals, Faculty of Medicine, Beni-Suef University, Beni-Suef 62521, Egypt; 3Department of Clinical Pharmacy, Faculty of Pharmacy, Nahda University (NUB), Beni-Suef 62521, Egypt; 4Department of Pharmacy Practice, College of Pharmacy, Princess Nourah bint Abdulrahman University, P.O. Box 84428, Riyadh 11671, Saudi Arabia; aysaleh@pnu.edu.sa; 5Department of Pharmaceutical Sciences, College of Pharmacy, Princess Nourah bint Abdulrahman University, P.O. Box 84428, Riyadh 11671, Saudi Arabia; asali@pnu.edu.sa; 6Clinical Pharmacy Department, Faculty of Pharmacy, Beni-Suef University, Beni-Suef 62521, Egypt; mohamed.abdelrahim@pharm.bsu.edu.eg

**Keywords:** impulsivity, anxiety, depression, functional impairment, COVID-19, pandemic

## Abstract

*Background and Objectives:* It is well known that depression, anxiety, and impulsiveness are interrelated; however, studies that have assessed their association with the coronavirus outbreak are scarce. Hence, our study aimed to evaluate the impulsivity incidence and its correlation with anxiety and depression following COVID-19 infection between November 2022 and June 2023. *Materials and Methods:* The 201 participants completed the Hamilton Depression Rating Scale (HDRS), Hamilton Anxiety Rating Scale (HAM-A), Sheehan Disability Scale (SDS), and short UPPS-P scales (urgency, premeditation, perseverance, sensation seeking, and positive urgency) to allow us to determine their anxiety and depression symptoms, functional impairment, and impulsivity, respectively. *Results:* Among our respondents, 22%, and 26.7% had moderate to severe anxiety and depression. The short UPPS-P scale significantly correlated with the HAM-A and HDRS scales. Participants with positive COVID-19 infection showed significantly higher functional impairment scores, especially in the work/study domain (mean (SD): 3.12 (2.2) vs. 2.43 (2.3); *p* = 0.037). COVID-19-related disruption significantly correlated with negative and positive urgency, HAM-A, HDRS, and the SDS total and subscales. *Conclusions:* Our findings showed a notable increase in anxiety, depression, and functional impairment among the population following COVID-19 infection. Our research highlights the correlation between impulsivity and the psychological distress experienced following the pandemic.

## 1. Introduction

The global outbreak of COVID-19, caused by severe acute respiratory syndrome coronavirus-2 (SARS-CoV-2) [1,2], dramatically affected human health, with unprecedented challenges to mental health [3,4]. Robust evidence has directly linked the COVID-19 pandemic with multiple psychological disorders [5,6,7]. However, there is still a paucity of data regarding the incidence or prevalence of these psychological disorders before and during the pandemic. The pandemic caused a condition of future uncertainty and panic amongst the population, whether fear of infection, losing family members, or losing jobs or economic stability. During the outbreak’s initial phase, several research studies investigated the psychological impact and reported a prevalence of more than 30% of the general population having had moderate to severe levels of anxiety [8,9,10]. In addition, community-based epidemiological studies documented high rates of depression, anxiety, and suicidal thoughts as a consequence of the pandemic [11,12,13]. A previous Spanish study showed that suicidal thoughts increased, with the highest rates in 2021 [14]. There were substantial changes in emergency visits related to suicidal thoughts between 2019, 2020, and 2021 (*p* < 0.001). The data revealed a notable rise in prevalence in 2021, particularly during the spring and summer seasons (3.2% and 4.2%, respectively), in comparison with previous years [14]. Similarly, Elhawary et al. conducted a study in Egypt that revealed a significant rise in hospital admissions caused by suicidal ideation or self-poisoning incidents from March 2019 to February 2021. Adolescents experienced a substantial rise, from 34.6% to 41.7%. During the second wave of the pandemic, there was a significant increase in the proportion of suicide deaths, rising from 1.9% to 21.2%. Admissions to the intensive care unit for poisoning and suicides significantly increased throughout the outbreak (*p* < 0.001) [15].

During the pandemic, health crises led to psychological changes, not solely in medical workers but also in citizens, instigated by fear, insecurity, anxiety, or depression [16,17,18]. The identification of individuals in the early stages of a psychological disorder makes intervention strategies more effective. Impulsivity is a multidimensional construct and a fundamental feature of human behavior; however, this common problem has public health consequences [19,20]. Impulsivity can be particularly interrelated with anxiety and depression. Moreover, it has been reported that the presence of anxiety increases impulsive attitudes such as suicidal ideation and attempts [21]. While not everyone with anxiety engages in suicidal behavior, impulsivity reduces self-restraint, leading some individuals to seek quick and easy escapes from stressful situations. This highlights a potential connection between stress and impulsivity. Since therapy can be a source of stress, this lack of self-restraint may lead to evasion and non-adherence to therapeutic or precautionary measures in clinical settings during an outbreak. Social isolation and lack of connection heighten the risk of deteriorating mental health, suggesting that preventive health measures like self-quarantining and social distancing may have adverse effects on mental well-being [22,23].

Indeed, two-thirds of individuals in the general population reported at least some anxiety or depression manifestations during the initial phases of the pandemic. However, these estimates also appeared to decrease globally as the pandemic became considerably prolonged [24]. Nonetheless, there is a paucity of studies investigating the impact of emotion-related impulsivity and its linkage with anxiety and depression in response to the pandemic. Therefore, we aimed to assess the prevalence of impulsivity and its association with anxiety and depression in the normal population during and after the COVID-19 pandemic.

## 2. Materials and Methods

### 2.1. Subjects and Procedures

This was a cross-sectional study based on an online survey that was carried out in seven months from November 2022 to June 2023. We recruited participants from the Beni-Suef Governorate general population in Egypt.

The sample size needed from the population was estimated using the following formula:$$n0=\frac{Z^{2}\times p\times q}{e^{2}}$$
where *Z* = 1.96 for a 95% CI, *e* = margin of error = 0.05, *p* = 0.5, and *q* = 1 − *p*. The estimated sample size (*n*0) ~ 384. The modified Cochran formula is used for small populations’ sample calculation:$$n0=n0/(1+\frac{n0-1}{N}\;)$$

*N* = population, *n*0 = 384/(1 + (384 − 1/93) = 75.1 = 75. Therefore, a minimum recommended sample size was set at 75 participants using G*power software 3.1 to achieve a power level of 80% and alpha error of 5% [17,18]. Via random sampling, adult participants aged between 18 and 60 years who were able to consent to this study and respond to the questionnaires were included in the analysis. We excluded participants aged less than 18, those with psychiatric or medical illnesses that would interfere with the diagnostic assessment, and those who did not adequately complete the questionnaires. The survey was conducted via the Google Forms tool, which necessitates that participants log into the platform using an email account to take part in the survey, preventing duplicate entries from a single account [25].

The questionnaire was disseminated using several social media sites, email, and conventional messaging services. A brief description of the main purpose and procedure of this study was included at the beginning of the survey, followed by an online informed consent form. The current study commenced after approval from the Faculty of Medicine/Beni-Suef University (FMBSU) research ethical committee (REC) (FMBSUREC/06112022).

### 2.2. Study Measures

The questionnaire had two main parts: first, demographic data collection, and second, the following scales:

#### 2.2.1. The Arab Short Urgency–Premeditation–Perseverance–Sensation-Seeking—Positive Urgency Impulsive Behavior Scale (Short UPPS-P) [26]

For impulsivity assessment, we utilized a short UPPS-P scale, which consists of 20 items translated into Arabic with five subscales to assess impulsivity: negative urgency (NU), lack of premeditation (LP), lack of perseverance (LoP), positive urgency (PU), and sensation seeking (SS) [26]. For each item, respondents rated the extent of agreement or disagreement with statements describing general attitudes by which people think or act. Response options on a 4-point scale—strongly agree, agree, disagree, and strongly disagree—were scored as 1 to 4, respectively, where higher scores denoted higher impulsivity traits. The subscales had satisfactory internal consistency, as evidenced by Cronbach’s α values ranging from 0.734 to 0.770.

#### 2.2.2. Hamilton Anxiety Rating Scale (HAM-A) [27]

There were 14 statements for anxiety assessment that described certain conditions or feelings that people may have. Five options were included to gauge the extent to which the respondent experienced these conditions, ranging from not present to very severe. The total score ranged from 0 to 56, with a score of less than 17 indicating mild anxiety, 18 to 24 denoting mild to moderate anxiety, and 25 to 30 moderate to severe anxiety (Cronbach’s α = 0.728).

#### 2.2.3. Hamilton Depression Rating Scale (HDRS) [28]

This consisted of 17 items with four varying responses to assess depression; participants were asked to select the one amongst them that best described their condition [28] A score of up to 7 was considered normal, while a score of ≥20 indicated moderate depression. The estimated Cronbach’s α calculated to assess the questionnaire’s validity was 0.761.

#### 2.2.4. COVID-19-Related Disruption

A 12-item checklist was used to assess the extent to which each participant experienced COVID-19-related disruption. We used items in multiple domains, including illness exposure (whether they or anyone they knew had tested positive), employment (became unemployed due to the pandemic or believed there was over a 50% likelihood that they would lose their job due to the pandemic), worsened financial situation, and social disruption (including items concerning diminished quality of social interactions). For each item answered “yes”, a score of 1 was given, and the total score denoted the total number of disruption events experienced [24]. The internal validity of the score was satisfactory, as denoted by Cronbach’s alpha (α = 0.69).

#### 2.2.5. The Sheehan Disability Scale (SDS) [29]

This scale was used to gauge functional impairment in three primary interrelated domains (work/study performance, social life, and household/family life). Potential participants were asked to choose the worst month for them in the past 6 months in terms of mental health manifestations and then to rate the interference of these symptoms as follows: 0, not at all; 1–3, mild impairment; 4–6, moderately impaired; 7–9, markedly impaired; and 10, extremely impaired. The possible overall score was up to 30. The final two items asked participants about the total duration for which their symptoms led them to miss school and/or work or be unproductive [30]. The internal consistency as evidenced by Cronbach’s α value, which ranged from 0.795 to 0.826, in the three domains was adequate.

### 2.3. Statistical Analysis

We conducted statistical analyses to assess the socio-demographic characteristics of the respondents. We used the mean and standard deviation (SD) to express numerical data. We made comparisons between respondents who had previously contracted COVID-19 and those who had not. We assessed data normality using the Kolmogrov–Smirnov test. The psychometric scale scores, such as UPPS-s, HAM-A, HDRS, and SDS, exhibited a normal distribution. Therefore, we used the mean (SD) in this data presentation. An unpaired Student *t*-test was used for quantitative data, while categorical variables were analyzed using the Chi-square test. We conducted univariate comparisons between individuals with and without a verified COVID-19 infection. We used Spearman’s correlation between impulsivity, anxiety, depression, and functional impairment as indicated by the Sheehan Disability Scale to identify the association between these variables. We performed linear regression analysis to determine the effects of impulsivity on HAM-A, HDRS, SDS, and COVID-19-associated disruptions. We estimated the beta-coefficient, *p*-values, and 95% confidence intervals (CIs), and we evaluated the overall fit of the model using R2 and F-tests. Since the missing data in our retrieved responses were minimal, we conducted a complete case analysis. We set the significance level at a 2-sided *p*-value of approximately 0.05. All analyses were conducted using SPSS IBM version 23.0.

## 3. Results

### 3.1. Socio-Demographic Characteristics

The overall response rate among invited participants was 42.8% (201/469). Throughout the timeframe of the trial, a total of 201 participants adequately completed the survey and were included in the analyses. Fifteen participants were excluded from the study as they did not adequately complete the questionnaires. The majority of participants were female (70.1%), and 29.9% were male. The mean ± SD age of the recruited participants was 39.51 ± 16.07, with a range of between 18 and 60. Of them, 14.9% were married (Table 1).

### 3.2. COVID-19 Pandemic-Related Disruption

The majority of respondents (89%, n = 179) reported disruption events associated with COVID-19 in at least one of the assessed domains including exposure to illness and occupational, financial, or social-related events, and about 68% (n = 137) reported disruption in multiple domains. Specifically, 54.2% (n = 109) of participants were infected with COVID-19, and 62.7% (n = 126) had a family member who tested positive. During the pandemic, 11.4% (n = 23) were made unemployed, while 110 participants (54.7%) did not lose their jobs. Approximately 18.4% of the employed participants believed there was over a 50% risk they would become unemployed due to COVID-19, while 21.9% (n = 44) believed that the pandemic made it harder for them to pay their rent/mortgage. Social interactions had worsened in 50.7% (n = 102), as reported by the respondents, and 61.2% (n = 123) reported poor concentration.

### 3.3. Psychometric Scales for Anxiety, Depression, and Functional Impairment

The mean ± SD score of HAM-A among all participants was 15.46 ± 10.17. According to the HAM-A scale, 50.7% experienced mild anxiety, 27.3% experienced mild to moderate anxiety, and 22% of respondents were in the moderate to severe anxiety category. Female participants showed higher anxiety levels compared to males and the difference was statistically significant (17.51 ± 10.33 vs. 10.6 ± 7.97, *p* < 0.001). The mean ± SD Hamilton Depression Rating score was 13.89 ± 8.56 for all participants, with 54 (26.7%) experiencing moderate depression, which comprised mostly of females (83.3%, n = 45), and the difference in HDRS for depression was statistically significant (*p* = 0.005), as shown in Table 2. We also observed consistently higher HAM-A and HDRS scores among those who reported positive COVID-19 infection compared to those who denied previous infection, with mean ± SD values of 16.61 ± 9.95 vs. 14.09 ± 10.32 and 14.79 ± 9.12 vs. 12.82 ± 7.76, respectively. However, the difference was not statistically significant (*p* = 0.079 and 0.104). A statistically significantly high score of HAM-A and HDRS was also observed among respondents who reported a previous history of depression, panic or a state of fear, and poor concentration (*p* < 0.001).

Using the Sheehan disability scale, the assessment of functional impairment in the study population showed mean ± SD scores of 2.81 ± 2.3 for the work/study domain, 3.12 ± 2.5 for the social domain, and 3.4 ± 2.8 for the family life domain. The frequencies of SDS domains are summarized in Figure 1.

Participants with a positive infection with COVID-19 showed consistently higher scores of functional impairment. However, the effect was significant in the work/study domain only (*p* = 0.037). Participants who experienced a history of panic or fear showed significantly higher SDS scores; however, this effect reached significance only in the family domain (4.04 ± 2.9 vs. 2.83 ± 2.5; *p* = 0.002). Participants who reported a history of depression showed consistently higher SDS scores in the three domains, and the difference was statistically significant [(3.28 ± 2.5 vs. 2.23 ± 1.9, *p* = 0.001), (3.61 ± 2.7 vs. 2.54 ± 2.3, *p* = 0.003), and (3.85 ± 2.9 vs. 2.85 ± 2.6, *p* = 0.011), for the work/study domain, social life domain, and family domain, respectively]. Similarly, participants who reported a condition of poor concentration showed consistently significantly higher scores in the three main scale items, as follows: 3.3 ± 2.4 vs. 2.03 ± 1.9, *p* < 0.001 for the work subscale, 3.57 ± 2.6 vs. 2.42 ± 2.3, *p* = 0.002 for the social subscale, and 3.91 ± 2.8 vs. 2.59 ± 2.6, *p* = 0.001 for the family subscale.

In the main scale items, females consistently scored higher than males. However, the difference was statistically significant in only two domains as follows: 3.06 ± 2.4 vs. 2.2 ± 1.8, *p* = 0.015 for the work subscale, and 3.4 ± 2.6 vs. 2.47 ± 2.2, *p* = 0.017 for the social subscale (Table 3).

### 3.4. UPPS

The total score and subscale of impulsiveness are summarized in Table 4. The mean ± SD scores in the five domains of the short-UPPS scale were 8.89 ± 2.52, 7.99 ± 2.03, 8.33 ± 2.25, 8.26 ± 2.24, and 9.51 ± 2.71 for the NU, LP, LoP, PU, and SS, respectively, with an overall score of 42.97 ± 8.49. The total scores of impulsivity were higher in respondents who reported positive COVID-19 infection (43.41 ± 7.12) compared to the non-infected respondents (42.45 ± 9.88); however, the difference was not significant (*p* = 0.422). The same also applied to the impulsivity subscales; whereas consistently higher scores were observed among respondents who tested positive for COVID-19 without a significant difference (*p* = 0.597, 0.461, 0.890, 0.694, and 0.304 for negative urgency, lack of premeditation, lack of perseverance, positive urgency, and sensation seeking, respectively).

### 3.5. Correlation and Regression Analyses

We conducted Spearman’s rank Rho correlation analysis to evaluate the link between the impulsive behavior scale and the outcomes of the other psychometric scales including HAM-A, HDRS, and SDS. Table 5 summarizes Spearman’s rank correlations between the outcome variables. A significant inter-correlation between the impulsivity traits was observed. The five traits of impulsivity (positive and negative urgency, sensation seeking, lack of perseverance, and premeditation) significantly correlated with the HAM-A and HDRS scales. Also, the SDS, HAM-A, and HDRS total scores were interrelated as demonstrated by the statistically significant correlations (r = 0.484, *p* < 0.001, and r = 0.509, *p* < 0.001, respectively). COVID-19-related disruption showed a significant correlation with two domains of impulsivity (negative and positive urgency), HAM-A, HDRS, and the SDS total and subscales.

Regression analysis to identify the predictors of the overall s-UPPS scale was performed, and the results are summarized in Table 6. After adjustment for the socio-demographic confounders, three predictive dimensions were identified for the impulsivity overall score, including HAM-A (β = 0.307, *p* = 0.004), HDRS (β = 0.146, *p* = 0.016), and COVID-19-related disruption (β = 1.024, *p* < 0.001), with R^2^ = 0.416, and F = 7.920.

## 4. Discussion

The current study used an online survey to highlight the psychological consequences of the outbreak, including depression, anxiety, and its association with functional impairment and impulsivity post COVID-19 pandemic. Our findings are congruent with the existing literature concerning the psychological effects of the COVID-19 outbreak on a worldwide scale [31,32]. During the pandemic waves, there was a notable escalation in stress and anxiety levels among the general population [33,34]. The long-term consequences of COVID-19 have been adequately described in the literature. While most studies focused on the clinical sequelae of the infection, few studies assessed the protracted psychological disorders associated with the pandemic. Our HAM-A and HDRS score results revealed increased anxiety and depression levels among the participants following the COVID-19 outbreak. Furthermore, these scores significantly increased with is the patient has a history of infection. These outcomes are consistent with those reported by Poyraz et al., who reported substantially high psychological distress, anxiety, and depression as protracted symptoms following recovery from acute COVID-19 infection [35]. Similarly, in a study conducted by Liu et al. (2020), the prevalence levels of moderate-to-severe depression and anxiety were approximately 10% and 20%, respectively, within around one month following hospital discharge [36]. Infection with the virus has been implicated in the onset of psychological disorders according to animal models and post-mortem brain analysis of infected cases. The proposed underlying mechanism is the potential for viral penetration of the blood–brain barrier, triggering immune responses that in turn have been related to mood and psychological disturbances [37].

Even though multiple studies have documented various psychological distresses linked to the COVID-19 pandemic, only a few studies have specifically examined the functional impairment experienced by individuals during this era. Over 20% of our respondents reported moderate functional impairment, as represented by the SDS. It is noteworthy that we observed consistently higher SDS scores among respondents with previous COVID-19 infection; however, the difference was only significant in the work/study domain. We also found a significant correlation between the trajectory of functional impairment and COVID-19 disruption events. Wilson et al. (2022) also described disruptions in the social, work, and family domains, reporting that 11% of those who tested positive for the virus experienced moderate to marked impairment in any category of the SDS and at least one moderate to severe symptom lasting for a minimum of 8 months. In comparison, just 2% of the participants who tested negative for the virus reported similar levels of impairment and symptoms, and the relative risk (RR) for experiencing these outcomes was 4.5 [95% CI: 2.7 to 7.3] [34]. Married people may have experienced greater concerns during the pandemic related to the psychological and physical status of their family members, spouse, and children [38]. Despite the relatively low percentage of married people compared to unmarried ones (14.9% vs. 85.1%) among our study sample, we observed high scores of SDS among married participants; however, the difference in scores did not reach significance. These findings are consistent with the findings of Park et al., 2022, who reported a robust association between COVID-19-related functional impairment, and being married with a high monthly income [39]. Another predictor of increased functional impairment was a prior history of depression or psychiatric illness. The current study population who reported a history of depression, infection panic, or fear showed significantly higher functional impairment scores. Moreover, a significant association with HAM-A and HDRS scores was observed with the degree of functional impairment. Individuals who suffer from depression often exhibit a decline in their level of productivity, a diminished ability to engage in interpersonal contact, and a sense of disconnection from social networks [40].

The value of using of impulsivity as a comprehensive risk factor for several clinical issues remains undisputed. While there is substantial evidence linking emotion-related impulsivity to internalizing symptoms, there is a paucity of research examining the association between this attribute and stress reactivity. We observed a significant association between anxiety/depression and impulsivity, which corroborates previous findings from the literature [41,42].

In our study, COVID-19-related disruption was significantly associated with negative and positive urgency. Negative and positive urgency have attracted increased interest as predisposing factors for internalizing disorders. Several research studies have reported their robust association with psychological distress and anxiety [43,44]. Pautrat et al.’s (2022) study conducted during the COVID-19 lockdown showed that these two traits of impulsivity were associated with emotion dysregulation, as well as the increased tendency to react impulsively in response to emotional stimuli [44]. To explain the links of impulsivity with stress, depression, and anxiety, it has been hypothesized that stressful life events can lead to impulsivity with worsened decision-making and coping abilities [45]. Individuals with high or low emotional-related impulsivity, while sharing the same stressful situations or life events, tend to differ in their coping strategies with these situations [44]. An increased tendency to use alcohol and its related problems were observed in subjects with high negative urgency levels for coping purposes [46]. In other words, increased stress could be conducive to health risk behaviors, such as suicidal ideation and substance use mediated by increased impulsivity [47]. In addition, findings from observational studies showed the linkage between impulsivity and risk-taking attitudes or lack of obedience with the restrictive or precautionary measures during the pandemic, with more risk of psychological disorders or emotional disruption post COVID-19 recovery [48,49].

Gender also seems to have a remarkable role in the extent of psychological effects associated with the pandemic. Females showed significantly higher depressive and anxiety scores, which is consistent with the existing evidence from the literature. We also explored the possible relationship between gender and impulsivity. Among our participants, females showed higher impulsivity overall scores compared to males; however, a significant difference was only observed in the sensation-seeking domain. These findings seem to be contradictory to the common data from the literature [50]. However, in agreement with our findings, a few reports showed that women exhibit a higher tendency to make impulsive decisions compared to males, and the ultimate degree of impulsivity is contingent upon the specific tasks and subject samples involved [51]. Johnson et al. (2013) reported high impulsivity in females, in addition to the interplay of multiple factors such as neurotransmitters, particularly the serotonergic biomarker [52]. It is important to note that the interpretation of this finding should be approached with caution, as our study sample consisted of a higher proportion of females (70.1%) compared to males (29.9%).

Overall, although most studies from the literature focused on the psychological consequences during the outbreak, our study highlights the protracted depression, anxiety, and increased functional impairment post-pandemic. Our findings also emphasize the association between impulsivity and the psychological burden following the pandemic. Therefore, psychological support and education in managing negative emotions may help decrease the negative consequences associated with increased impulsivity and psychological distress. Given the ease of assessing emotion-related impulsivity, there is the potential to conduct broader screenings for these traits, hence creating an avenue for early interventions [52]. Furthermore, it may be beneficial to incorporate instruction on emotion recognition, self-regulation techniques for managing emotional states, and proactive coping strategies into conventional stress reduction methods employed in mental health interventions, such as mindfulness practices and social connection promotion techniques [51,52].

Despite the potential benefits of our findings to public health, this study has some limitations. The major limitations are the dependence on self-report questionnaires, and relying on responses from online social media platforms, which might cause selection bias by excluding potential participants with limited social media activity or those without internet access. To ensure manageability, we made an effort to minimize the length of the survey. Consequently, certain baseline parameters, such as body mass index (BMI), which necessitates self-measurement, were not gathered. While we inquired about the overall health condition before COVID-19, we did not specifically determine the occurrence or absence of each symptom that was mentioned, and we could not collect the pre-pandemic baseline data of participants. Due to the variation in severity levels among respondents during the illness, there was a potential for recall bias in this study, as the data regarding the functional status and disability was obtained retrospectively. Another limitation is that females made up the majority of our sample. It is worth noting that the tendency of women to participate in surveys more frequently than males has been recorded in previous publications as well, potentially because of personality or gender role variations [53,54]. As a result, the ability to extrapolate our findings to the male population is limited. In future research, it is recommended that researchers expand the participant sample size and incorporate behavioral assessments to validate the associations between impulsivity, anxiety, and depression.

## 5. Conclusions

Our study emphasizes the high prevalence of anxiety and depression following the pandemic, significantly associated with functional impairment. A prior history of COVID-19 infection and psychological disorders such as depression, fear, or panic, in addition to the increased disruption events associated with the pandemic, are risk factors for increased depression, anxiety, and consequent functional impairment. In addition, our results raise concern regarding the association of increased impulsivity with a previous history of COVID-19 infection. Therefore, it seems that identifying negative emotions and impulsivity can help in targeting people and fine-tuning interventions such as regular psychological support and education on negative emotion management to mitigate the harmful consequences of the pandemic. Future longitudinal studies are necessary with adequate follow-up of participants, to assess the persistence of impulsive behaviors following recovery from COVID-19 infection.

## Figures and Tables

**Figure 1 medicina-60-01367-f001:**
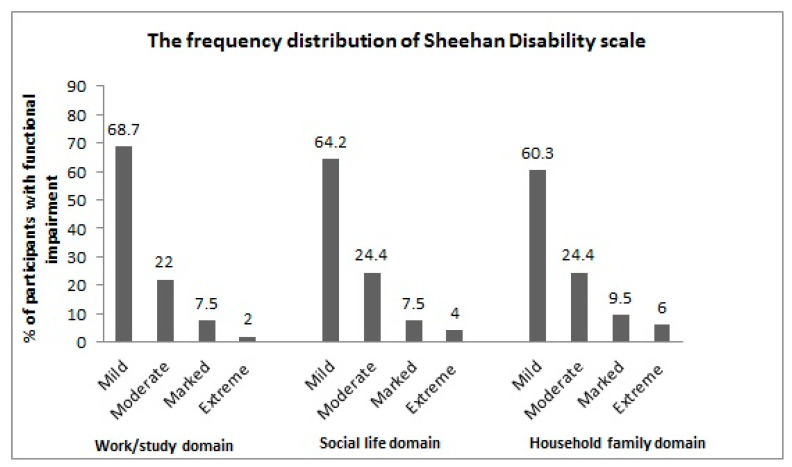
The frequency distribution of the Sheehan Disability Scale results in our study participants.

**Table 1 medicina-60-01367-t001:** Participants’ main characteristics.

Characteristic	N, %
1. Gender (n, %)	
Male	61, 29.9%
Female	140, 70.1%
2. Marital state (n, %)	
Single	171, 85.1%
Married	30, 14.9%
3. COVID-19-related disruption (yes, %)	
Positive infection with coronavirus	109, 54.2%
Family member infection with the coronavirus	126, 62.7%
Hospital admission as a result of infection with the coronavirus	13, 6.5%
Family member death as a result of corona infection	33, 16.4%
History or currently suffers from depression	110, 54.7%
History or currently suffers from a state of fear and panic	94, 46.8%
Suffers from obsessive–compulsive disorder	48, 23.9%
Became unemployed since the pandemic	23, 11.4%
Possibility of becoming unemployed due to the pandemic	37, 18.4%
Corona-related events affect the ability to pay rent or mortgage	44, 21.9%
Feeling that there are people available to talk to about problems (including online)	102, 50.7%
Suffers from poor concentration	123, 61.2%

**Table 2 medicina-60-01367-t002:** Scores of anxiety and depression scales in the sample population.

Key Variables	HAM-A		HDRS	
	Mean (SD)	*p*-Value	Mean (SD)	*p*-Value
Overall	15.46 (10.17)	<0.001 *	13.89 (8.56)	0.005 *
Male (29.9%)	10.6 (7.97)		11.28 (8.04)	
Female (70.14%)	17.51 (10.33)		14.99 (8.57)	
Married (%)		0.03 *		0.306
Yes (14.9)	19.17 (10.98)		15.37 (8.43)	
No (85.1)	14.81 (9.91)		13.63 (8.59)	
COVID-19 infection (%)		0.079		0.104
Yes (54.2)	16.61 ± 9.95		14.79 (9.12)	
No (45.8)	14.09 ± 10.32		12.82 (7.76)	
History of panic or fear state (%)		<0.001 *		<0.001 *
Yes (46.8)	19.36 (9.61)		16.78 (8.39)	
No (53.2)	12.03 (9.42)		11.35 (7.92)	
History of depression (%)		<0.001 *		<0.001 *
Yes (54.7)	18.86 (9.72)		16.55 (8.52)	
No (45.3)	11.34 (9.17)		10.63 (7.46)	
Poor concentration (%)		<0.001 *		<0.001 *
Yes (61.2)	19.05 (9.72)		16.61 (8.28)	
No (38.8)	9.79 (8.12)		9.59 (7.15)	

* indicates significance. HAM-A: Hamilton Anxiety Rating Scale (HAM-A), HDRS: Hamilton Depression Rating Scale (HDRS).

**Table 3 medicina-60-01367-t003:** Sheehan disability scores across the three main domains in the population sample.

	Sheehan Disability Scale					
	Work/Study Domain		Social Life Domain		Family Domain	
	Mean (SD)	*p*-Value	Mean (SD)	*p*-Value	Mean (SD)	*p*-Value
Overall	2.81 (2.3)	**0.015**	3.12 (2.5)	**0.017**	3.40 (2.7)	0.079
Male (29.9%)	2.2 (1.8)		2.47 (2.2)		2.87 (2.7)	
Female (70.14%)	3.06 (2.4)		3.40 (2.6)		3.62 (2.8)	
Married (%)		0.620		0.833		0.829
Yes (14.9)	3.0 (1.9)		3.03 (2.2)		3.5 (2.6)	
No (85.1)	2.77 (2.3)		3.14 (2.6)		3.38 (2.8)	
COVID-19 infection (%)		**0.037**		0.175		0.214
Yes (54.2)	3.12 (2.2)		3.35 (2.6)		3.62 (2.7)	
No (45.8)	2.43 (2.3)		2.86 (2.5)		3.13 (2.9)	
History of panic or fear state (%)		**0.006**		0.073		**0.002**
Yes (46.8)	3.29 (2.6)		3.47 (2.6)		4.04 (2.9)	
No (53.2)	2.38 (2.0)		2.82 (2.4)		2.83 (2.5)	
History of depression (%)		**0.001**		**0.003**		**0.011**
Yes (54.7)	3.28 (2.5)		3.61 (2.7)		3.85 (2.9)	
No (45.3)	2.23 (1.9)		2.54 (2.3)		2.85 (2.6)	
Poor concentration (%)		**<0.001**		**0.002**		**0.001**
Yes (61.2)	3.3 (2.4)		3.57 (2.6)		3.91 (2.8)	
No (38.8)	2.03 (1.9)		2.42 (2.3)		2.59 (2.6)	

Significance represented as bold.

**Table 4 medicina-60-01367-t004:** Participants’ impulsivity profile.

S-UPPS	Mean (SD)	Male Mean (SD)	Female Mean (SD)	*p*-Value
Total	42.97 (8.49)	40.77 (9.26)	43.91 (7.99)	**0.016**
Negative urgency	8.89 (2.52)	8.37 (2.59)	9.11 (2.47)	0.057
Lack of premeditation	7.99 (2.03)	7.67 (2.04)	8.12 (2.02)	0.148
Lack of perseverance	8.33 (2.25)	7.97 (2.18)	8.48 (2.27)	0.138
Positive urgency	8.26 (2.24)	8.03 (2.09)	8.36 (2.30)	0.343
Sensation seeking	9.51 (2.71)	8.73 (2.83)	9.84 (2.59)	**0.008**

S-UPPS: Short Urgency–Premeditation–Perseverance–Sensation-Seeking—Positive Urgency-Impulsive Behavior Scale. Significance represented as bold.

**Table 5 medicina-60-01367-t005:** Correlation between psychometric measures and key variables.

	1	2	3	4	5	6	7	8	9	10	11	12	13	14
1. S-UPPS total	1													
r														
P														
2. Negative urgency		1												
r	0.684													
P	**<0.001**													
3. Lack of premeditation			1											
r	0.507	**0.120**												
P	**<0.001**	**0.042**												
4. Lack of perseverance				1										
r	0.634	0.267	0.519											
P	**<0.001**	**<0.001**	**<0.001**											
5. Positive urgency					1									
r	0.695	0.574	0.140	0.340										
P	**<0.001**	**<0.001**	**0.048**	**<0.001**										
6. Sensation seeking						1								
r	0.689	0.421	0.208	0.295	0.408									
P	**<0.001**	**<0.001**	**0.003**	**<0.001**	**<0.001**									
7. HAM-A score							1							
r	0.247	0.260	0.308	0.264	0.178	0.159								
P	**<0.001**	**0.003**	**<0.001**	**0.001**	**0.011**	0.024								
8. HDRS score								1						
r	0.226	0.163	0.289	0.253	0.177	0.171	0.764							
P	**<0.001**	**0.018**	**<0.001**	**<0.001**	**0.012**	0.015	**<0.001**							
9. SDS work/study domain									1					
r	0.125	0.020	0.158	0.046	−0.018	0.047	0.478	0.449						
P	0.077	0.779	**0.020**	0.516	0.800	0.344	**<0.001**	**<0.001**						
10. SDS social domain										1				
r	0.157	0.031	**0.164**	0.116	**0.142**	0.076	0.447	0.505	0.687					
P	**0.026**	0.661	**0.020**	0.110	**0.044**	0.281	**<0.001**	**<0.001**	**<0.001**					
11. Family domain											1			
r	0.072	0.013	0.114	−0.002	−0.020	0.080	0.395	0.425	0.578	0.690				
P	0.307	0.986	0.108	0.982	0.783	0.260	**<0.001**	**<0.001**	**<0.001**	**<0.001**				
12. SDS total												1		
r	0.110	0.022	**0.144**	0.041	0.052	0.084	0.484	0.509	0.814	0.902	0.874			
P	0.122	0.753	**0.042**	0.563	0.466	0.238	**<0.001**	**<0.001**	**<0.001**	**<0.001**	**<0.001**			
13. Gender													1	
r	0.141	0.124	0.091	0.103	0.066	0.176	0.312	0.209	0.166	0.202	**0.167**	0.200		
P	**0.045**	0.068	0.201	0.147	0.354	**0.012**	**<0.001**	**0.003**	**0.018**	**0.004**	**0.018**	**0.004**		
14. COVID-19-related disruption														1
r	0.082	0.145	0.044	0.083	0.374	0.052	0.480	0.497	0.391	0.355	0.372	0.416	0.078	
p	0.442	**0.038**	0.539	0.241	**0.036**	0.461	**<0.001**	**<0.001**	**<0.001**	**<0.001**	**<0.001**	**<0.001**	0.272	

Significance represented as bold.

**Table 6 medicina-60-01367-t006:** Linear regression analysis for predictors of impulsivity.

Predictors	B	95% CI		*p*
HAM-A score	0.307	0.082	0.430	**0.004**
HDRS	0.146	0.057	0.435	**0.016**
SDS	−0.010	−0.204	0.184	0.092
COVID-19-related disruption	1.024	1.511	0.536	**<0.001**

CI: confidence interval. Significance represented as bold.

## Data Availability

The datasets used and/or analyzed during the current study are available from the corresponding author on reasonable request.

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
