# Peer review of "Impulsivity and Its Association with Depression and Anxiety in the Normal Egyptian Population Post COVID-19 Pandemic"

_medicina, 2024, doi:10.3390/medicina60081367_

Round 1

Reviewer 1 Report

Comments and Suggestions for Authors

Dear Authors,

I have reviewed the manuscript entitled "Impulsivity and its Association with Depression and Anxiety in The Normal Egyptian Population Post COVID-19 Pandemic." Regarding the current global scenario, the paper's subject matter is very relevant. Still, there are no major critical issues that would lead me to accept the paper for publication.

### Key Issues

1. **Insufficient Comparison Pre- and Post-Pandemic** The manuscript did not show a robust relationship between COVID-related disruption and psychological measures: negative and positive urgency, HAM-A, HDRS, and SDS total and subscales. All data were obtained during the pandemic; hence, there is no pre-pandemic baseline against which one could compare any change.

It is due to such a reason that there is simply no previous baseline existing before the pandemic against which inferences of observed presentations of increases in indices of anxiety, depression, and functional impairment can justifiably be made.

2. **COVID-19 Disruption Section Validation **

In the case of COVID-related disruption, it does not appear that the authors have developed it with any existing scientific validation. The reliability and validity of this variable raise a severe question concerning the validity of the findings.

3. ** Analysis and Conclusions

The tests performed would appear to be too weak to underpin the inferences in the paper; the absence of a pre-pandemic dataset for comparison and the measure of COVID-19-related disruption not being validated both entirely weaken the strength of evidence due to the relationships reported.

Thus, an impact of the pandemic on anxiety and depression levels and generalized functional impairment could be concluded, but without any valid interpretation.

Conclusion Addressing these critical methodological and analytical weaknesses, the manuscript cannot at this moment achieve the threshold value necessary for its publication. 

Author Response

**Insufficient Comparison Pre- and Post-Pandemic** The manuscript did not show a robust relationship between COVID-related disruption and psychological measures: negative and positive urgency, HAM-A, HDRS, and SDS total and subscales. All data were obtained during the pandemic; hence, there is no pre-pandemic baseline against which one could compare any change.

It is due to such a reason that there is simply no previous baseline existing before the pandemic against which inferences of observed presentations of increases in indices of anxiety, depression, and functional impairment can justifiably be made.

Author: All participants were healthy and showed no signs of illness with depression, or anxiety before contracting COVID-19. They only reported these issues after becoming infected. Therefore, we concluded that these disruptions are linked to the COVID-19 infection. Thank You

  1. **COVID-19 Disruption Section Validation **

In the case of COVID-related disruption, it does not appear that the authors have developed it with any existing scientific validation. The reliability and validity of this variable raise a severe question concerning the validity of the findings.

Author: Thanks for your comment. Indeed the internal validity of this developed questionnaire is crucial for the findings to be scientifically valid. We have already evaluated the internal reliability and validity of this questionnaire as and the score was satisfactory as denoted by Cronbach alpha (α=0.69).

  1. ** Analysis and Conclusions

The tests performed would appear to be too weak to underpin the inferences in the paper; the absence of a pre-pandemic dataset for comparison and the measure of COVID-19-related disruption not being validated both entirely weaken the strength of evidence due to the relationships reported. Thus, an impact of the pandemic on anxiety and depression levels and generalized functional impairment could be concluded, but without any valid interpretation.

Author: The pandemic's effect on anxiety and depression levels is evident, as all participants were healthy and showed no signs of illness, depression, or anxiety before contracting COVID-19. They only reported these issues after infection. Thus, we concluded that these disruptions are linked to COVID-19.Thank You

Conclusion Addressing these critical methodological and analytical weaknesses, the manuscript cannot at this moment achieve the threshold value necessary for its publication. 

Author: we assessed in this study, the prevalence of impulsivity and its association with anxiety and depression in the normal population post-COVID-19 pandemic as a dangerous outcomes of the infection, this study point is important as these disruptions may affect the work life and the economics of the person and the community. We have summarized our findings in the conclusion section.

Reviewer 2 Report

Comments and Suggestions for Authors

The manuscript is really interesting and well-written. However, it requires a major revision to improve it. The comments are the following:

- in the abstract, it is not clear this part of the text "[...]  following the COVID [...]" (row 23): do this part mention patients who had COVID or the pandemic period? Please, revise it;

- in the abstract, it would be better reporting the number of subjects potentially participant (row 24) and the total number of responders (row 27);

- in the abstract, there is no the full name of the acronym UPPS-P (row 26). Please, revise it;

- in the abstract, there are no space and bracket in this part of the text (row 30) " [...] (mean (SD):3.12(2.2) vs. 2.43(2.3); P=0.037). Please, revise it;

- in the abstract, the capital letter after the full stop is mandatory in this part of the text (row 32) "[...] . our findings showed a notable [...]". Please, revise it;

- in the abstract (rows 31-32) it would be better reporting the main results;

- in the introduction (rows 47-48), it would be better reporting the rates of the mental outcomes and the related geographical areas;

- the rows 52-53 "Impulsivity is a multidimensional [...]" represent a new paragraph. Please, start a new paragraph after the part of the text (row 52) "[...]  the intervention strategies more effective";

- this part of the phrase (rows 57-59) "the presence of impulsivity lowers the inner constraint and encourages people to choose quick and simple ways to get out of stressful situations" seems to be unclear: what are these simple ways to get out of stressful situations? Please, clarify;

- in the rows 59-60, it is not well explained the link between stress and impulsivity. Please, revise it and go into detail;

- the phrase in the rows 60-62 is not intelligible: why could therapy be a potential source of stress? Could the COVID pandemic play a key role in this? Also could the social distancing have an impact on population's psychological condition? Please, clarify;

- The role of social distancing in this introduction is lacking. Please, add something;

- As reported above, it is not clear how impulsivity is increased during the pandemic. For what reasons did the population develop such impulsivity during COVID pandemic? Please, clarify;

- the phrase in the rows 64-65 is not clear: why did these estimates appear to decrease? What are the reasons? Please, clarify;

- in the row 69, considering the study period, writing "post-COVID-19 pandemic" seems to be uncorrect because the pandemic started on 11th March 2020 and finished on 5th May 2023. I suggest writing "during and after COVID pandemic";

- in the row 72, writing better the study design chosen: I suggest "cross-sectional study based on an online survey/questionnaire";

- in the row 73, it is not clear where Beni-Suef Governorate is (what country?). Please, clarify;

- in the rows 75-77, writing better the phrase, putting the reference before the colon (row 75), writing the formula better going to the new line as a classical fraction (numerator at the top and denominator at the bottom of the fraction line), and adding the values of n0 and N in the formula, remembering that the minimum sample size for cross-sectional study changes according to the prevalence of the outcome of interest (depression and anxiety in this case). Moreover, add the citation or the references of the G*power software;

- in the rows 77-78, it is not clear how the casual sampling of the general population was done. Please, go into detail;

- in the row 81, how was the diagnostic assessment done? Is the questionnaire self-administrated or during the medical examination? Please, clarify this aspect;

- in the row 82 "[...] which were 15 responses in total" must be in the results section. Please, revise it;

- the citation of Google Forms, mentioned in the rows 82-83, is lacking. Please, add it;

- in the rows 86-87, it is no clear who the guidelines are for. Please, clarify;

- in the rows 89-92, it is mentioned the anonymity of the questionnaire, but you ask the potential participants to fill the questionnaire after log in the own email account. This could comprimise the anonymity of the questionnaire. Therefore, how did you face this aspect? How did you mantain the anonymity if the potential participants had to fill the questionnaire entering in the own email account, with the possibility to give and link the own email in the questionnaire? Please, clarify this aspect going into detail;

- in the row 92, there are more brackets and full stops than usual " [...] committee of (FMBSUREC/06112022).).". Moreover, what is the meaning of this acronym FMBSUREC?

- in the rows 94-95, it is not clear what questions are included in the demographic section of the questionnaire (age? sex? residence?). Please, clarify;

- in the row 95, describe better the second section of the questionnaire, giving a little mention of the scales used. Moreover, remove the colum at the end of the sentence;

- in the rows 96-104, there is no cut off of impulsivity. Moreover, it is not clear whether the questionnaire is validated or not and there is no Cronbach's alpha. Please, revise it;

- in the rows 105-110, there is no reference of this questionnaire. Moreover, it is not clear whether the questionnaire is validated or not and there is no Cronbach's alpha. Please, revise it;

- in the rows 111-114, it is not clear whether the questionnaire is validated or not and there is no Cronbach's alpha. Please, revise it;

- in the rows 115-124, there is no reference of this questionnaire. Moreover, it is not clear whether it is a questionnaire validated previously and there is no Cronbach's alpha. Please, revise it;

- in the rows 125-133, it is not clear whether it is a questionnaire validated previously and there is no Cronbach's alpha. Please, revise it;

- in the rows 137-138, it is not clear how you conducted the comparison "[...] between respondents who reported prior infection with COVID-19 and those without a history of infection." What statistical tests did you use to detect it? Please, revise it;

- in the rows 138-139, it is important to notice that a normality test must be performed before choosing the measure of central tendency (mean for normally distributed data or median for the non-normally). Moreover, before the normality test is performed, then it is chosen the best measure of central tendency. Please, rewrite better;

- in the row 139, student t-test is for normally distributed data. Please, add the alternative test for non-normally distributed data;

- in the rows 141-142, it is not clear how to verify the subjects who had COVID-19. Moreover, this criterion is not mentioned above, when the study population is described. Please, clarify and go into detail;

- in the row 142, even if the missing data is minimal, this must be reported in the results section, specifying which variables have missing data. Moreover, it is important to define minimal missing data (is there a cut off for this?);

- in the row 143, a correlation is mentioned. Which correlation? Please, define it better and go into detail for variables correlated;

- in the rows 145-146, there is the criterion on subjects with previous COVID. Please, add this in the section where the study population is described and where inclusion and exclusion criteria are listed;

- in the row 147, it seems to be an incomplete citation of the statistical software and unnecessary punctuation signs: "[...] (Install Shield Corporation, Inc.,).". Please, revise it;

- in the row 150, the response rate is not mentioned. Considering the number of potential participants (written in the materials and methods section), please add it;

- please, choose the best measure of central tendency after performing the normality test or explain better the choice of using mean and SD in statistical analyses section;

- please, formatting better the Table 1 in order to have the variables in one row or rename the variables with shorter way. Moreover, revise the numbered list of variables (choosing if the number must be followed by full stop or middle dash for the same variables list);

- in row 158, it seems to be an unnecessary bracket: "[...] financial, or social-related events) [...]". Please, revise it;

- in the row 160, how was SARS-CoV-2 positivity detected in relatives? Was it reported by the participants or was there a medical report of such positivity? Please, clarify and mention this aspect in materials and methods;

- in the rows 168-182, the paragraph needs to have a deeper revision in terms of acronyms, considering the previous full name with the related acronym between brackets for the first time (after that, you can use the acronym in the remaining part of the manuscript). Please, revise it;

- in the row 175, the Table 2 was mentioned in the middle of the paragraph. In this way, it is not intelligible which table do the remaining results belong to. Please, put the Table 2 either at the beginning or at the end of the paragraph;

- in the row 179, there is no the P indicating the p-value: "[...] not statistically significant (P=0.079, and 0.104)." Please, revise it;

- in the rows 186-188, the results are not intelligible. Please, rewrite them clearer;

- in the rows 189-191, the three domains should be defined better, not reporting at the end of the sentence because the readability of the sentence becomes more difficult. Please, revise it;

- please, improve the image quality of Figure 1;

- in the row 195, it would be better to divide the sentence, putting a full stop before "[...] however" and change the first letter into capital. Please, revise;

- in the row 197, it would be better to change "[...] however [...]" with "but";

- in the rows 198-200, it would be better to enclose the p-value into brackets in order to improve the readability. Please, revise it;

- please, revise the bold in Table 2 and Table 3, using it for only the table header;

- in the row 202, it would be better to have the full name of the title of the paragraph instead of the related acronym. Please, revise it;

- in the row 203, there is another synonym of impulsivity. Please, choose only one and use it for the entire manuscript;

- in the rows 203-206, it would be better to rewrite the sentence because of its poor intelligibility, especially understanding what are the five domains. Please, clarify;

- in the row 209, revise the semi-colon before "[...] whereas [...]", please;

- in the row 210-212, revise the sentence making it more intelligible. I could suggest putting the related p-value into brackets after the defined domain; Moreover, pay attention for the bracket at the end of the sentence "[...] urgency and sensation seeking, respectively).";

- in the Table 4, there are two statistically significant associations (total and sensation seeking). Please, report these results into the related section. Moreover, simplify the related section (rows 202-212) highlighting only the statistically significant results;

- the rows 217-219 must be put in the statistical analyses section;

- in the rows 225-227, there are not the results of the correlations. Please, add them;

- in the row 227, start a new paragraph after "[...] scales.";

- in the rows 228-229, there are not the variables considered as confounders. Please, mention them in the statistical analyses section;

- please, put the all statistically significant results in the Tables of the results section in bold;

- please, write something on the regression analysis and the type of regression analysis used in the statistical analysis section of the manuscript, going into detail on measures reported (beta coefficient or odds ratio, confidence intervals and so on) and related adjustments (for example, the adjustment for sex and age);

- in the row 230, there are some mistakes in terms of missing spaces or unnecessary spaces, as well as the way in which COVID was written, which are highlighted in yellow here "[...] including HAM-A (β=0.307,P=0.004), HDRS(β=0.146, P=0.016 ), and Covid-19-related". Please, correct them;

- in the row 231, there is this part of the text "[...] with R2=0.416, and F=7.920." In statistica analyses section there is no mention of these measures. Please, add them and go into detail;

- in the Table 5, it is important to put the variable in a single row in order to improve the readability. Please, revise it;

- in the discussion section, it is important to improve the comparison between your results and the literature. Please, implement it, adding some epidemiological data;

- in the discussion section, it would be better to explain the biological basis of the mental outcomes detected in this study, going into detail about the role of SARS-CoV-2 in worsening of mental outcomes. Please, improve this aspect;

- in the rows 258-260, some references seem to be lacking. Please, revise and add them, repeating them if you start a new paragraph;

- in the rows 261-262, it is important to revise the acronym and the related full name: "[...] Sheehan Disability Scale. It is noteworthy that we observed consistently higher SDS scores [...]". Please, do it;

- in the rows 267-274, some references seem to be lacking. Please, revise and add them, repeating them if you start a new paragraph;

- in the row 272, it would be better to start a new paragraph after the reference number 25;

- in the rows 278-279, there are not results of the association reported. Please, implement them;

- in the row 279, it would be better to start a new paragraph after the reference number 27;

- in the rows 279-286, there is no comparison between your results and the literature. Please, implement this paragraph with measures of association or other epidemiological data;

- in the rows 287-292, there is no comparison between your results and the literature. Please, implement this paragraph with measures of association or other epidemiological data, as well as adding biological basis of this aspect;

-  in the rows 304-311, there is no comparison between your results and the literature. Please, implement this paragraph with some epidemiological data, adding the comparison between subjects with alcohol abuse or other coping mechanisms and impulsivity and subject with the same coping mechanisms without impulsivity. The same must be done for other mental outcomes, with or without impulsivity;

- in the row 312-318, there is no explanation about the biological mechanisms in females with impulsivity and other conditions which can explain the impulsivity among females. Please, implement this aspect;

- in the row 323, there is too many spaces between words: "[...] biomarker (40). It is important [...]". Please, revise it;

- in the rows 326-328, there is no epidemiological data on the mental outcomes before and during the pandemic. Please, implement this aspect and add something also in the introduction section;

- even if the questionnaire is anonymous, social desirability bias might be present in this study. Please, assess if this bias might be present in this study and explain how to face it;

- the future directions and policies of this study are not define well. Please, go into detail about how this study implement the current policies on population's mental health in terms of prevention and treatment, and give some directions for future pandemic scenario in order to prevent mental outcomes.

Author Response

The manuscript is really interesting and well-written. However, it requires a major revision to improve it. The comments are the following:

- in the abstract, it is not clear this part of the text "[...]  following the COVID [...]" (row 23): do this part mention patients who had COVID or the pandemic period? Please, revise it;

Author: Done, Thank You

- In the abstract, it would be better reporting the number of subjects potentially participant (row 24) and the total number of responders (row 27);

Author: Done, Thank You

- in the abstract, there is no the full name of the acronym UPPS-P (row 26). Please, revise it;

Author: Done, Thank You

- in the abstract, there are no space and bracket in this part of the text (row 30) " [...] (mean (SD):3.12(2.2) vs. 2.43(2.3); P=0.037). Please, revise it;

Author: Done, Thank You

- in the abstract, the capital letter after the full stop is mandatory in this part of the text (row 32) "[...] . our findings showed a notable [...]". Please, revise it;

Author: Done, Thank You

- in the abstract (rows 31-32) it would be better reporting the main results;

Author: Done, Thank You

- in the introduction (rows 47-48), it would be better reporting the rates of the mental outcomes and the related geographical areas;

Author: Added, Thank You

- the rows 52-53 "Impulsivity is a multidimensional [...]" represent a new paragraph. Please, start a new paragraph after the part of the text (row 52) "[...]  the intervention strategies more effective";

Author: Done, Thank You

- this part of the phrase (rows 57-59) "the presence of impulsivity lowers the inner constraint and encourages people to choose quick and simple ways to get out of stressful situations" seems to be unclear: what are these simple ways to get out of stressful situations? Please, clarify;

Author: Done, Thank You

- in the rows 59-60, it is not well explained the link between stress and impulsivity. Please, revise it and go into detail;

Author: Done, Thank You

- the phrase in the rows 60-62 is not intelligible: why could therapy be a potential source of stress? Could the COVID pandemic play a key role in this? Also could the social distancing have an impact on population's psychological condition? Please, clarify;

Author: clarified, Thank You

- The role of social distancing in this introduction is lacking. Please, add something;

Author: added, Thank You

- As reported above, it is not clear how impulsivity is increased during the pandemic. For what reasons did the population develop such impulsivity during COVID pandemic? Please, clarify;

Author: clarified, Thank You

- the phrase in the rows 64-65 is not clear: why did these estimates appear to decrease? What are the reasons? Please, clarify;

Author: It decrease as the pandemic dissipate and quarantine has become non-existent.

- in the row 69, considering the study period, writing "post-COVID-19 pandemic" seems to be uncorrect because the pandemic started on 11th March 2020 and finished on 5th May 2023. I suggest writing "during and after COVID pandemic";

Author: Corrected, Thank You

- in the row 72, writing better the study design chosen: I suggest "cross-sectional study based on an online survey/questionnaire";

Author: Corrected, Thank You

- in the row 73, it is not clear where Beni-Suef Governorate is (what country?). Please, clarify;

Author: clarified, Thank You

- in the rows 75-77, writing better the phrase, putting the reference before the colon (row 75), writing the formula better going to the new line as a classical fraction (numerator at the top and denominator at the bottom of the fraction line), and adding the values of n0 and N in the formula, remembering that the minimum sample size for cross-sectional study changes according to the prevalence of the outcome of interest (depression and anxiety in this case). Moreover, add the citation or the references of the G*power software;

Author: Done, Thank You

- in the rows 77-78, it is not clear how the casual sampling of the general population was done. Please, go into detail;

Author: Done, Thank You

- in the row 81, how was the diagnostic assessment done? Is the questionnaire self-administrated or during the medical examination? Please, clarify this aspect;

Author: The survey was conducted via the Google Forms tool, which necessitates participants to log into the platform using an email account to take part in the survey, hence preventing duplicate entries from a single account. The questionnaire was disseminated using several social media sites, email, and conventional messaging services.

- in the row 82 "[...] which were 15 responses in total" must be in the results section. Please, revise it;

Author: These responses were excluded as they did not adequately complete the questionnaires

- the citation of Google Forms, mentioned in the rows 82-83, is lacking. Please, add it;

Author: added, Thank You

- in the rows 86-87, it is no clear who the guidelines are for. Please, clarify;

Author: Corrected, Thank You

- in the rows 89-92, it is mentioned the anonymity of the questionnaire, but you ask the potential participants to fill the questionnaire after log in the own email account. This could comprimise the anonymity of the questionnaire. Therefore, how did you face this aspect? How did you mantain the anonymity if the potential participants had to fill the questionnaire entering in the own email account, with the possibility to give and link the own email in the questionnaire? Please, clarify this aspect going into detail;

Author: this part was corrected, Thank You

- in the row 92, there are more brackets and full stops than usual " [...] committee of (FMBSUREC/06112022).).". Moreover, what is the meaning of this acronym FMBSUREC?

Author: the approval was obtained from a committee of faculty of medicine Beni-Suef university (FMBSU)

- in the rows 94-95, it is not clear what questions are included in the demographic section of the questionnaire (age? sex? residence?). Please, clarify;

Author: that was illustrated in table 1, thank you

- in the row 95, describe better the second section of the questionnaire, giving a little mention of the scales used. Moreover, remove the colum at the end of the sentence;

Author: I think that is important to describe the scales for the young readers, thank you

- in the rows 96-104, there is no cut off of impulsivity. Moreover, it is not clear whether the questionnaire is validated or not and there is no Cronbach's alpha. Please, revise it;

Author: Thanks for your comment. Indeed the internal validity of this developed questionnaire is crucial for the findings to be scientifically valid. We have already evaluated the internal reliability and validity of this questionnaire as and the score was satisfactory as denoted by Cronbach alpha (α=0.69).

- in the rows 105-110, there is no reference of this questionnaire. Moreover, it is not clear whether the questionnaire is validated or not and there is no Cronbach's alpha. Please, revise it;

Author: added, Thank You

- in the rows 111-114, it is not clear whether the questionnaire is validated or not and there is no Cronbach's alpha. Please, revise it;

Author: We have already evaluated the internal reliability and validity of this questionnaire as and the score was satisfactory as denoted by Cronbach alpha (α=0.69).

- in the rows 115-124, there is no reference of this questionnaire. Moreover, it is not clear whether it is a questionnaire validated previously and there is no Cronbach's alpha. Please, revise it;

Author: added, Thank You

- in the rows 125-133, it is not clear whether it is a questionnaire validated previously and there is no Cronbach's alpha. Please, revise it;

Author: We have already evaluated the internal reliability and validity of this questionnaire as and the score was satisfactory as denoted by Cronbach alpha (α=0.69).

- in the rows 137-138, it is not clear how you conducted the comparison "[...] between respondents who reported prior infection with COVID-19 and those without a history of infection." What statistical tests did you use to detect it? Please, revise it;

Author: revised, Thank You

- in the rows 138-139, it is important to notice that a normality test must be performed before choosing the measure of central tendency (mean for normally distributed data or median for the non-normally). Moreover, before the normality test is performed, then it is chosen the best measure of central tendency. Please, rewrite better;

Author: done, Thank You

- in the row 139, student t-test is for normally distributed data. Please, add the alternative test for non-normally distributed data;

Author: Clarified, Thank you

- in the rows 141-142, it is not clear how to verify the subjects who had COVID-19. Moreover, this criterion is not mentioned above, when the study population is described. Please, clarify and go into detail;

Author: during the survey, the patients demonstrated the COVID-19 history, thank you

- in the row 142, even if the missing data is minimal, this must be reported in the results section, specifying which variables have missing data. Moreover, it is important to define minimal missing data (is there a cut off for this?);

Author: the results are only for the participants who completed the survey adequately, not including the missing data

- in the row 143, a correlation is mentioned. Which correlation? Please, define it better and go into detail for variables correlated;

Author: The correlation are illustrated in table 5

- in the rows 145-146, there is the criterion on subjects with previous COVID. Please, add this in the section where the study population is described and where inclusion and exclusion criteria are listed;

Author: clarified, thank you

- in the row 147, it seems to be an incomplete citation of the statistical software and unnecessary punctuation signs: "[...] (Install Shield Corporation, Inc.,).". Please, revise it;

Author: revised, Thank You

- in the row 150, the response rate is not mentioned. Considering the number of potential participants (written in the materials and methods section), please add it;

Author: corrected, thank you

- please, choose the best measure of central tendency after performing the normality test or explain better the choice of using mean and SD in statistical analyses section;

Author: clarified, thank you

- please, formatting better the Table 1 in order to have the variables in one row or rename the variables with shorter way. Moreover, revise the numbered list of variables (choosing if the number must be followed by full stop or middle dash for the same variables list);

Author: revised, Thank You

- in row 158, it seems to be an unnecessary bracket: "[...] financial, or social-related events) [...]". Please, revise it;

Author: revised, Thank You

- in the row 160, how was SARS-CoV-2 positivity detected in relatives? Was it reported by the participants or was there a medical report of such positivity? Please, clarify and mention this aspect in materials and methods;

Author: That was reported by the participants as all the study was via an online survey

- in the rows 168-182, the paragraph needs to have a deeper revision in terms of acronyms, considering the previous full name with the related acronym between brackets for the first time (after that, you can use the acronym in the remaining part of the manuscript). Please, revise it;

Author: That was reported in methodology section

- in the row 175, the Table 2 was mentioned in the middle of the paragraph. In this way, it is not intelligible which table do the remaining results belong to. Please, put the Table 2 either at the beginning or at the end of the paragraph;

Author: revised, thank you

- in the row 179, there is no the P indicating the p-value: "[...] not statistically significant (P=0.079, and 0.104)." Please, revise it;

Author: revised, thank you

Round 2

Reviewer 1 Report

Comments and Suggestions for Authors

The authors did not improve the paper.

It is not acceptable in the present form

Author Response

I have reviewed the manuscript entitled "Impulsivity and its Association with Depression and Anxiety in The Normal Egyptian Population Post COVID-19 Pandemic." Regarding the current global scenario, the paper's subject matter is very relevant. Still, there are no major critical issues that would lead me to accept the paper for publication.

**Insufficient Comparison Pre- and Post-Pandemic** The manuscript did not show a robust relationship between COVID-related disruption and psychological measures: negative and positive urgency, HAM-A, HDRS, and SDS total and subscales. All data were obtained during the pandemic; hence, there is no pre-pandemic baseline against which one could compare any change.

It is due to such a reason that there is simply no previous baseline existing before the pandemic against which inferences of observed presentations of increases in indices of anxiety, depression, and functional impairment can justifiably be made.

Answer: Thank you for the comment. Indeed, we agree with the reviewer that the presence of baseline data before pandemic would provide more robust evidence on the deleterious consequences of infection with COVID-19. However the findings of this study are still important. Since, we compared those infected with COVID-19 versus respondents who denied previous infection with the virus. Interestingly, our findings revealed higher anxiety, depression, and functional impairment, and consequently, higher impulsivity. Impulsiveness was also correlated with the assessed psychological abnormalities. It is noteworthy that there is shortage in Egypt regarding baseline data. So, we performed our study on the basis of self-report. This might be one of the limitations of our study and we have added this in the limitations section. However, the study take the precedence in assessment of impulsivity association with the disruption associated with the pandemic as one of the long-term consequences of the infection. Only few studies assessed the long term effects of COVID-19 infection and most of studies focus on the clinical symptoms post-infection. Moreover, this study might open the gate for further longitudinal studies with larger sample size.

  1. **COVID-19 Disruption Section Validation **

In the case of COVID-related disruption, it does not appear that the authors have developed it with any existing scientific validation. The reliability and validity of this variable raise a severe question concerning the validity of the findings.

Answer: Thanks for your comment. Indeed the internal validity of this developed questionnaire is crucial for the findings to be scientifically valid. We have already evaluated the internal reliability and validity of this questionnaire as and the score was satisfactory as denoted by Cronbach alpha (α=0.69). We have cited a reference for the questionnaire and performed internal validity assessment

  1. ** Analysis and Conclusions

The tests performed would appear to be too weak to underpin the inferences in the paper; the absence of a pre-pandemic dataset for comparison and the measure of COVID-19-related disruption not being validated both entirely weaken the strength of evidence due to the relationships reported. Thus, an impact of the pandemic on anxiety and depression levels and generalized functional impairment could be concluded, but without any valid interpretation.

Answer: Thank you for the comment. We compared those infected with COVID-19 versus respondents who denied previous infection with the virus. Interestingly, our findings revealed higher anxiety, depression, and functional impairment, and consequently, higher impulsivity. Impulsiveness was also correlated with the assessed psychological abnormalities. It is noteworthy that there is shortage in Egypt regarding baseline data. So, we performed our study on the basis of self-report. This might be one of the limitations of our study and we have added this in the limitations section. However, the study take the precedence in assessment of impulsivity association with the disruption associated with the pandemic as one of the long-term consequences of the infection. We also performed validation of COVID-19-related disruption measure.

Conclusion Addressing these critical methodological and analytical weaknesses, the manuscript cannot at this moment achieve the threshold value necessary for its publication. 

Answer: The study take the precedence in assessment of impulsivity association with the disruption associated with the pandemic as one of the long-term consequences of the infection. We also performed validation of COVID-19-related disruption measure. Only few studies assessed the long term effects of COVID-19 infection and most of studies focus on the clinical symptoms post-infection. Moreover, this study might open the gate for further longitudinal studies with larger sample size.

Reviewer 2 Report

Comments and Suggestions for Authors

I congratulated the authors for having improved their manuscript. However, some comments are still needed to implement it, listed below:

- in the lines 43-44, the phrase "Robust evidence has directly linked the COVID-19 pandemic 43 with multiple psychological disorders" is very impactful. However, incidence or prevalence of these psychological disorders before and during the pandemic are still lacking. Please, implement this part of the manuscript;

- in the lines 50-51, a Spanish study was correctely mentioned. If it is possibile, add an Egyptian study on the same issue. Furthermore, add the rates detected by the Spanish study, please;

- in the line 52, I suggest to remove "[...] for instance [...]" and rewrite the initial part of the sentence in this way (it is an example): "During the pandemic, health crises [...]";

- in the lines 83-84, it is appropriated to write the total population (total number) from which the random sample was extracted;

- in the lines 85-87, writing better the phrase, putting the reference before the colon (row 75), writing the formula better going to the new line as a classical fraction (numerator at the top and denominator at the bottom of the fraction line), and adding the values of n0 and N in the formula, remembering that the minimum sample size for cross-sectional study changes according to the prevalence of the outcome of interest (depression and anxiety in this case). Moreover, add the citation or the references of the G*power software;

- in the line 92, the subordinate "[...], which were 15 responses in total" must be removed because it is more appropriate to put it in the results section;

- in the line 93, the reference of Google Forms tool is still lacking. Please, add it;

- considering the lines 92-95, I would ask you how to ensure the anonymity of the questionnaire if there was a collection of participants' emails. On the contrary, if there is not a collection of participants' emails, specify it, please;

- in the lines 102-103, it is important to have the full meaning of the acronym FMBSUREC. Considering the previous answer "[...] committee of faculty of medicine Beni-Suef university (FMBSU)", please add it properly. Furthermore, revise the punctuation marks in this part: "[...] (FMBSUREC/06112022).).";

- considering the lines 105-106, what is the structurated questionnaire on demographic data? Please, specify the questions included. Moreover, it is appropriated to add the entire questionnaire in the Appendix;

- add, for each scales used, the Cronbach's alpha value, not only for COVID-related Disruption questionnaire, please;

- considering the lines 146-150, the normality test to check the values distribution of quantitative variables must be write before choosing the measures of central tendency (mean or median). After having written the normality test used, you can add the mean or median. I advice you it is better write that you choose mean or median after the result of normality test. On the contrary, if you choose mean regardless of the values distribution, remove the normality test. Please, revise it;

- it is strongly recommanded that Spearman’s rank Rho correlation analysis is added in the statistical analysis section. The same is for the regression analysis used (linear regression? logistic regression? other regression models?). Moreover, add the measures used such as confidence intervals, beta coefficent or odds ratio and so on;

- in the first part of the results section, a response rate (which is the ratio between the number of responders, the participants of this study, and the number of the population) must be added;

- in the line 171, remove the bracket in this part of the text: "[...] financial, or social-related events) [...]";

- please, remove the bold from the second column of the Table 2;

- in the lines 199-204, it is needed to revise this paragraph in this way (it is only a suggestion): "Using the Sheehan disability scale, the assessment of functional impairment in the study population showed mean±SD scores of 2.81±2.3 for the work/study domain, 3.12±2.5 for the social, and 3.4±2.8 for the family life. The frequency of SDS domains is summarized in Figure 1. A significantly higher SDS scores were observed among respondents with a history of depression for the work/study domain (P=0.001), the social (P=003), and the family life (P=0.011)." Furthemore, revise the p-value of the social domain (it was written 003, so it is impossible to understand if it is 0.003 or 0.03);

- in the line 128, add a space in this part of the text "[...] COVID-19related [...]", between 19 and related;

- in the line 208, I suggest to put a full stop before "[...] however, [...]". The same is for line 210;

- in the line 211, remove "[...] (work and social subscales) [...]";

- in the lines 211-213, I suggest to rewrite the part of the sentence in this way: "[...] 3.06±2.4 vs. 2.2±1.8; (P=0.015) for work subscale, 3.4±2.6 vs. 2.47±2.2; (P=0.017) for social subscale., and 3.62±2.8 vs. 2.87±2.7, P=0.079 for family life [...]", removing the last part, because the p-value is upper than 0.05;

- please, write the statistically significant results of the variables "History of panic or fear state", "History of depression", and "Poor concentration", which are in Table 3;

- please, remove the bold from the first column;

- please, revise the results in the section entitled "UPPS" because some results reported in the section are different from those in the Table 4;

- please, remove "descriptive" in the line 226;

- considering the lines 240-244, I suggest to put the sentence before the Table 6, not the Table 5;

- in the line 243, I suggest to revise this part of the sentence: "[...]  HAM-A (β=0.307,P=0.004), HDRS(β=0.146, P=0.016 ) [...]" because some spaces are lacking or too many spaces are between letter/number and puntuaction marks. Moreover, COVID must be in capital letter;

- in the lines 274-275, put the acronym of Sheehan Disability Scale, please;

- if you put the confidence intervals and the related acronym in the statistical analysis section for your results, it is not necessary use the full form in the line 285.

Comments on the Quality of English Language

The quality of English is good, but it might be improved. Some sentences are too long.

Author Response

Response to reviewers:

Dear editor, we have carefully reviewed all the comments and suggestions from reviewers. We would like to express our gratitude for the editor and reviewers for their excellent and insightful suggestions and taking time to review our manuscript. We provide a point by point author response below:

Reviewer 2

I congratulated the authors for having improved their manuscript. However, some comments are still needed to implement it, listed below:

Answer: We believe that our manuscript benefited a lot from your insightful comments and recommendations. Thank you for taking the time to help us improve our work. Please, find the response to the listed comments below point by point:

- in the lines 43-44, the phrase "Robust evidence has directly linked the COVID-19 pandemic 43 with multiple psychological disorders" is very impactful. However, incidence or prevalence of these psychological disorders before and during the pandemic are still lacking. Please, implement this part of the manuscript;

Answer: Has been added, thank you for insightful suggestion.

- In the lines 50-51, a Spanish study was correctely mentioned. If it is possibile, add an Egyptian study on the same issue. Furthermore, add the rates detected by the Spanish study, please;

Answer:

- in the line 52, I suggest to remove "[...] for instance [...]" and rewrite the initial part of the sentence in this way (it is an example): "During the pandemic, health crises [...]";

Answer: Thank you for your suggestion. The sentence has been edited.

- in the lines 83-84, it is appropriated to write the total population (total number) from which the random sample was extracted;

Answer: we used the following formula to estimate the sample size of unlimited population: n0 = Z2PQ/ e 2. The equation descried in the main text, Thank you.

- in the lines 85-87, writing better the phrase, putting the reference before the colon (row 75), writing the formula better going to the new line as a classical fraction (numerator at the top and denominator at the bottom of the fraction line), and adding the values of n0 and N in the formula, remembering that the minimum sample size for cross-sectional study changes according to the prevalence of the outcome of interest (depression and anxiety in this case). Moreover, add the citation or the references of the G*power software;

Answer: We have revised this sentence and added the reference

- in the line 92, the subordinate "[...], which were 15 responses in total" must be removed because it is more appropriate to put it in the results section;

Answer: Has been moved to the results section, Thank you.

- in the line 93, the reference of Google Forms tool is still lacking. Please, add it;

Answer: Has been added, Thank you.

- considering the lines 92-95, I would ask you how to ensure the anonymity of the questionnaire if there was a collection of participants' emails. On the contrary, if there is not a collection of participants' emails, specify it, please;

Answer: Thank you for your question. The collected emails didn’t necessarily refer to the participants’ identity. Most emails included numbers or acronyms not names. Moreover, the invitations disseminated through social media platforms. So. most responses received from foreign emails. By anonymity we meant that we didn’t include the name of respondents in the collected data and the statistical analysis was performed by specialist on the collected data. However, if this sentence seems inadequate, we have omitted it, Thank you.

- in the lines 102-103, it is important to have the full meaning of the acronym FMBSUREC. Considering the previous answer "[...] committee of faculty of medicine Beni-Suef university (FMBSU)", please add it properly. Furthermore, revise the punctuation marks in this part: "[...] (FMBSUREC/06112022).).";

Answer: the full meaning of FMBSUREC acronym has been added and punctuations corrected, Thank you.

- considering the lines 105-106, what is the structurated questionnaire on demographic data? Please, specify the questions included. Moreover, it is appropriated to add the entire questionnaire in the Appendix;

Answer: The collected demographic data included the gender and marital status. These data are presented in Table 1 and we omitted the word “structured”, Thank you.

- add, for each scales used, the Cronbach's alpha value, not only for COVID-related Disruption questionnaire, please;

Answer: Has been added, Thank you.

- considering the lines 146-150, the normality test to check the values distribution of quantitative variables must be write before choosing the measures of central tendency (mean or median). After having written the normality test used, you can add the mean or median. I advice you it is better write that you choose mean or median after the result of normality test. On the contrary, if you choose mean regardless of the values distribution, remove the normality test. Please, revise it;

Answer: Thank you for your recommendation. We have edited this sentence.

- it is strongly recommanded that Spearman’s rank Rho correlation analysis is added in the statistical analysis section. The same is for the regression analysis used (linear regression? logistic regression? other regression models?). Moreover, add the measures used such as confidence intervals, beta coefficent or odds ratio and so on;

Answer: the statistical analysis section has been edited as suggested, Thank you

- in the first part of the results section, a response rate (which is the ratio between the number of responders, the participants of this study, and the number of the population) must be added;

Answer: The response rate has been added in the first part of the results section, Thank you.

- in the line 171, remove the bracket in this part of the text: "[...] financial, or social-related events) [...]";

Answer: Has been omitted, Thank you.

- please, remove the bold from the second column of the Table 2;

Answer: Has been edited, Thank you.

- in the lines 199-204, it is needed to revise this paragraph in this way (it is only a suggestion): "Using the Sheehan disability scale, the assessment of functional impairment in the study population showed mean±SD scores of 2.81±2.3 for the work/study domain, 3.12±2.5 for the social, and 3.4±2.8 for the family life. The frequency of SDS domains is summarized in Figure 1. A significantly higher SDS scores were observed among respondents with a history of depression for the work/study domain (P=0.001), the social (P=003), and the family life (P=0.011)." Furthemore, revise the p-value of the social domain (it was written 003, so it is impossible to understand if it is 0.003 or 0.03);

Answer: Thank you for noticing the erroneous P-value. However, we have omitted this sentence based on the reviewer's upcoming suggestion, in which writing the statistically significant results of the variables "history of panic or fear state,"  "history of depression,"  and "poor concentration,"  which are in Table 3, was recommended.

- in the line 128, add a space in this part of the text "[...] COVID-19related [...]", between 19 and related;

Answer: Has been edited, Thank you.

- in the line 208, I suggest to put a full stop before "[...] however, [...]". The same is for line 210;

Answer: Has been edited, Thank you.

- in the line 211, remove "[...] (work and social subscales) [...]";

Answer: Has been omitted, Thank you.

- in the lines 211-213, I suggest to rewrite the part of the sentence in this way: "[...] 3.06±2.4 vs. 2.2±1.8; (P=0.015) for work subscale, 3.4±2.6 vs. 2.47±2.2; (P=0.017) for social subscale., and 3.62±2.8 vs. 2.87±2.7, P=0.079 for family life [...]", removing the last part, because the p-value is upper than 0.05;

Answer: Has been omitted, Thank you.

- please, write the statistically significant results of the variables "History of panic or fear state", "History of depression", and "Poor concentration", which are in Table 3;

Answer: Has been added, Thank you.

- please, remove the bold from the first column;

Answer: Has been removed, Thank you.

- please, revise the results in the section entitled "UPPS" because some results reported in the section are different from those in the Table 4;

Answer: Has been revised, Thank you.

- please, remove "descriptive" in the line 226;

Answer: Has been omitted, thank you.

- considering the lines 240-244, I suggest to put the sentence before the Table 6, not the Table 5;

Answer: we moved the sentences before Table 6 as suggested, Thank you.

- in the line 243, I suggest to revise this part of the sentence: "[...]  HAM-A (β=0.307,P=0.004), HDRS(β=0.146, P=0.016 ) [...]" because some spaces are lacking or too many spaces are between letter/number and puntuaction marks. Moreover, COVID must be in capital letter;

Answer: thank you for your recommendation. The sentence has been revised and corrected.

- in the lines 274-275, put the acronym of Sheehan Disability Scale, please;

Answer: Has been edited, thank you.

- if you put the confidence intervals and the related acronym in the statistical analysis section for your results, it is not necessary use the full form in the line 285.

Answer: has been edited, thank you for your notice

Comments on the Quality of English Language

The quality of English is good, but it might be improved. Some sentences are too long.

Answer: all manuscript has been revised regarding English language grammar and fluency.  We sincerely hope that we fulfilled the required edits and any further edits would be implemented as well. Thank you very much.

Round 3

Reviewer 1 Report

Comments and Suggestions for Authors

I have reviewed the manuscript entitled "Impulsivity and its Association with Depression and Anxiety in The Normal Egyptian Population Post COVID-19 Pandemic." Regarding the current global scenario, the paper's subject matter is very relevant. Still, there are no major critical issues that would lead me to accept the paper for publication.

Reviewer 2 Report

Comments and Suggestions for Authors

I am pleased to receive the manuscript again and I congratulate with the Authors for the excellent work. There are some mistakes in text editing, such as spaces between words and capital/small letters. However, after a properly text editing, the manuscript can be published.

My best wishes to the Authors.